# Image Compression Using Fractal Functions

Olga Svynchuk [1], Oleg Barabash [1], Joanna Nikodem [2], Roman Kochan [2] and Oleksandr Laptiev [3,*]

1. Department of Automation of Heat and Power Engineering Processes, National Technical University of Ukraine "Igor Sikorsky Kyiv Polytechnic Institute", 37 Peremohy Ave., 03056 Kyiv, Ukraine; 7011990@ukr.net (O.S.); bar64@ukr.net (O.B.)
2. Department of Computer Science and Automatics, University of Bielsko-Biala, 2 Willowa Str., 43-309 Bielsko-Biala, Poland; jnikodem@ath.bielsko.pl (J.N.); rkochan@ath.bielsko.pl (R.K.)
3. Department of Information and Cybersecurity Systems, State University of Telecommunications, 7 Solomenska Str., 03110 Kyiv, Ukraine
* Correspondence: alaptev64@ukr.net; Tel.: +380-674348001

**Abstract:** The rapid growth of geographic information technologies in the field of processing and analysis of spatial data has led to a significant increase in the role of geographic information systems in various fields of human activity. However, solving complex problems requires the use of large amounts of spatial data, efficient storage of data on on-board recording media and their transmission via communication channels. This leads to the need to create new effective methods of compression and data transmission of remote sensing of the Earth. The possibility of using fractal functions for image processing, which were transmitted via the satellite radio channel of a spacecraft, is considered. The information obtained by such a system is presented in the form of aerospace images that need to be processed and analyzed in order to obtain information about the objects that are displayed. An algorithm for constructing image encoding–decoding using a class of continuous functions that depend on a finite set of parameters and have fractal properties is investigated. The mathematical model used in fractal image compression is called a system of iterative functions. The encoding process is time consuming because it performs a large number of transformations and mathematical calculations. However, due to this, a high degree of image compression is achieved. This class of functions has an interesting property—knowing the initial sets of numbers, we can easily calculate the value of the function, but when the values of the function are known, it is very difficult to return the initial set of values, because there are a huge number of such combinations. Therefore, in order to de-encode the image, it is necessary to know fractal codes that will help to restore the raster image.

**Keywords:** fractal; fractal function; fractal properties; self-similarity; fractal compression; iterated function system; affine transformations





## 1. Introduction

Remote sensing of the Earth (REE) is the observation of our planet and the determination of the properties of objects on the earth's surface, obtained with the help of imaging devices installed on aircraft and artificial satellites of the Earth (Figure 1). The obtained data are aerospace images of the observed part of the earth's surface in digital form in the form of raster images (Figures 2 and 3). These data are subject to further detailed analysis, namely, the determination of quantitative characteristics of the object under study, which are necessary to predict the development of a phenomenon or process. The processing and interpretation of remote sensing data is closely related to digital image processing. Remote sensing allows one to obtain from space high-quality images of the earth's surface, which help to solve practical problems in various fields of human activity. However, the amount of this graphical information is very large and needs to be compressed when transmitting data over communication channels and then used in geographic information systems. Today, fractal methods are often used to transfer images from satellites to the ground, which help to give a better picture. The main feature of these methods is the property of

self-similarity of the image. Such methods provide large compression ratios. However, they need significant development while taking into account many criteria (speed, compression ratio, quality during decompression). They can be considered a real alternative to JPEG for many classes of images used in everyday life.

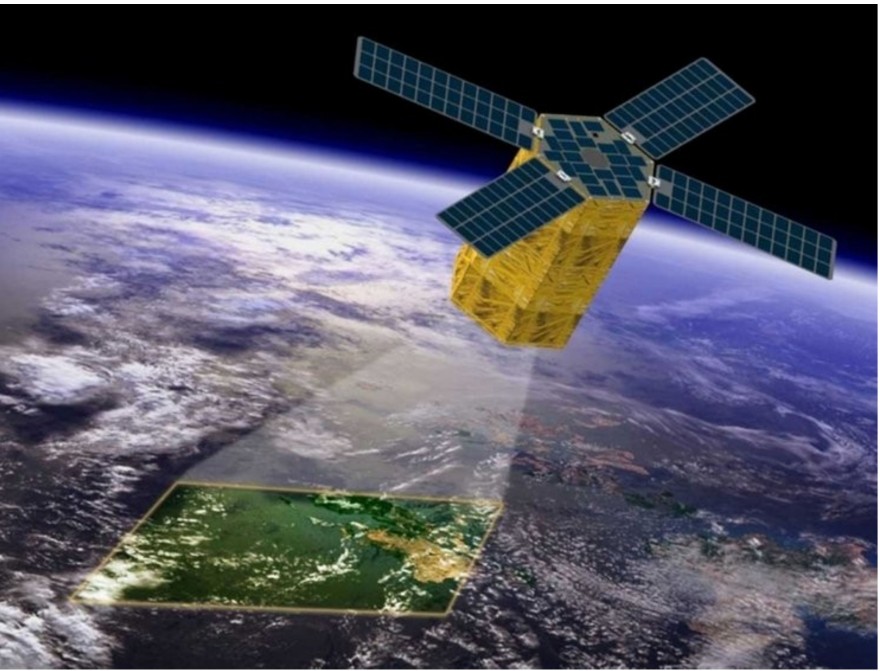

**Figure 1.** Remote sensing of the Earth.

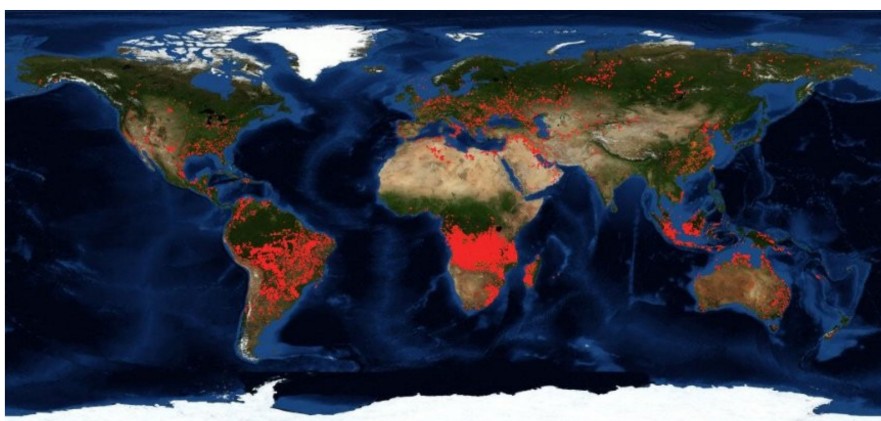

**Figure 2.** Fires on the planet Earth, 2020.

Fractal image compression (fractal transformation, fractal encoding) is a lossy image compression algorithm based on the iterated function system to images [1,2]. The encoding algorithm is encoded [3]. The concept of the fractal method was first introduced in 1990 by the British mathematician Michael Barnsley [4]. It consists of the fact that in the image it is necessary to find separate self-similar fragments which are repeated many times. He proved a class of theorems that allowed for the effective compression of images. In [5], Arno Jacquin described a new method of fractal encoding, in which the image is divided into domain and rank blocks, which cover the entire image. This approach became the basis for the creation of new fractal encoding methods used today. Jacquin's method was perfected by Yuval Fisher [6] and other scientists.

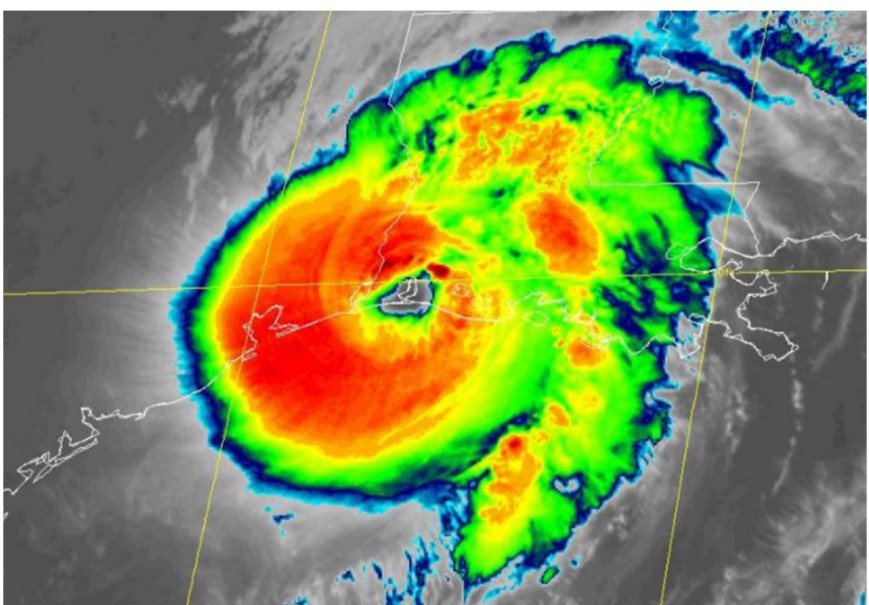

**Figure 3.** Hurricane Laura, USA, 2020.

A large number of scientific papers are devoted to the study of the effectiveness of the fractal image compression method [7–22]. For example, in [7] the algorithm of fractal compression of images using space-sensitive hashing is considered. Various methods are proposed in [8–13] aimed at reducing the volume of domain blocks, which makes it possible to reduce the encoding time of the search. In [8], a discrete cosine transform was used to classify all domain blocks into a number of classes, in [9] R-trees were used for this purpose, and in [10] a self-organized neural network was used. In [11,12], scientists use a genetic algorithm to search for optimal domains. In [13], the search is limited by the degree of information entropy of the domains. In [14–22], the variants of optimization and increase in fractal encoding speed and possibilities of their practical application are analyzed.

Today, the topic of fractal compression algorithms is still being actively studied.

The aim of our research is to model the class of fractal functions, study their properties and establish the possibility of their application for an efficient algorithm for encoding images that are presented in digital form.

## 2. Materials and Methods

Fractal encoding is a mathematical process for encoding rasters into a set of mathematical data that reproduces the fractal properties of a given image. This encoding is based on the fact that all natural objects have a lot of similar information in the form of repetitive patterns. They are called fractals. Fractal decoding is a reverse process in which a system of fractal codes is converted into a raster [14].

The concept of fractal was proposed by the French-American mathematician Benoit Mandelbrot. In 1977, he published "Fractal Geometry of Nature", describing repetitive drawings from everyday life [23]. According to him, many geometric figures consist of smaller figures, which when being enlarged repeat accurately a larger figure (Figures 4 and 5). After conducting research, he also found that fractals have chaotic behavior, a fractional infinite dimension (how completely a fractal fills a space when magnified to smaller details), and can be described mathematically using simple algorithms [24]. It is known that many geographical objects have fractal properties—contours of coasts and oceans, rivers, mountain gorges, and state borders where they are drawn by natural contours [24].

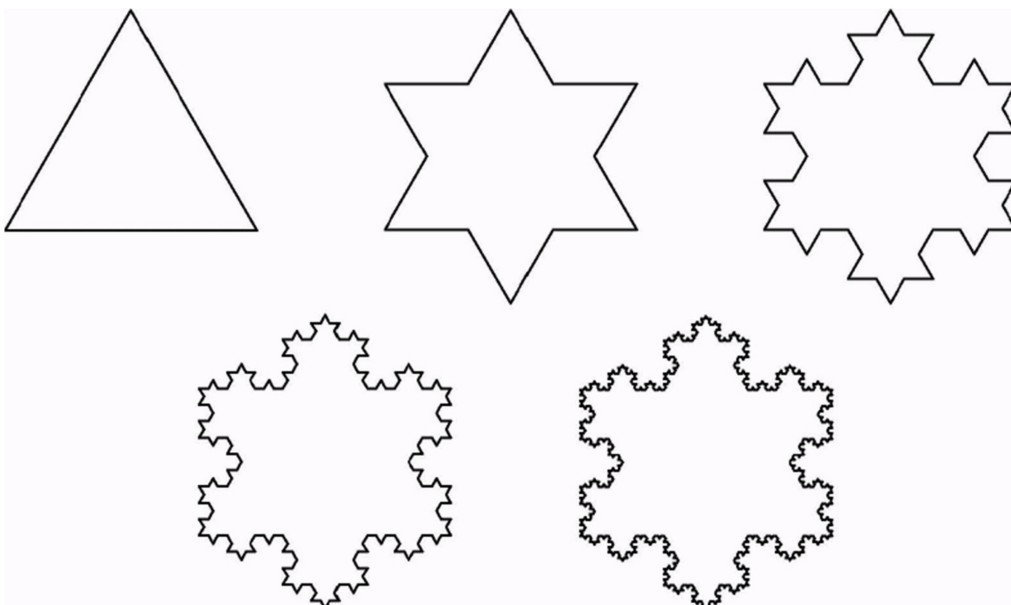

**Figure 4.** Koch snowflake.

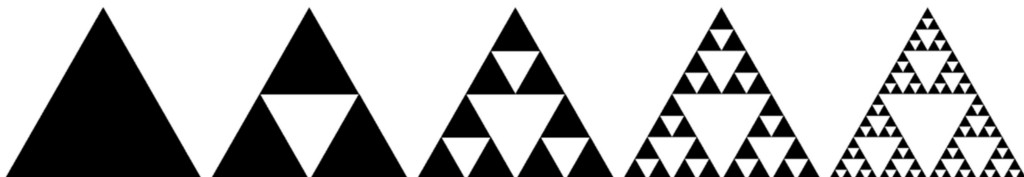

**Figure 5.** Sierpinski's triangle.

For the constructive task of such fractal objects and their analytical research today in mathematics, various systems of representation and the encoding of real numbers are widely applied: with the finite and infinite, constant and variable alphabet, with natural and the whole negative, rational and irrational bases, etc. These are s-s and non-s-s, Q-representations, chain fractions, real numbers in the series Cantor, Lurot, Engel, Sylvester, Ostrogradsky–Sierpinski–Pierce, etc [25–28]. The creation of a new encoding system for the fractional part of a real number significantly expands the range of such objects, which are relatively simply formally described and studied.

In our work, we use an encoding system with a finite alphabet to build a new mathematical model that will be used in fractal image compression. The model is based on a continuous class of continuous functions that depend on a finite set of parameters and have fractal properties.

Let $A_5 = \{0, 1, 2, 3, 4\}$ be the alphabet of the five-digit numeral system, $L \equiv A_5 \times A_5 \times A_5 \times \ldots A_5 \times \ldots$ be a space of the sequences of elements of the alphabet and let $Q_5^* = (q_{ij})$, $i \in A_5$, $j \in \mathbb{N}$ be an infinite stochastic matrix with positive elements $q_{ij} > 0$ and the following properties:

$$Q_5^* = (q_{ij}) = \begin{pmatrix} q_{01} & q_{02} & \cdots & q_{0j} & \cdots \\ q_{11} & q_{12} & \cdots & q_{1j} & \cdots \\ q_{21} & q_{22} & \cdots & q_{2j} & \cdots \\ q_{31} & q_{32} & \cdots & q_{3j} & \cdots \\ q_{41} & q_{42} & \cdots & q_{4j} & \cdots \end{pmatrix},$$

1.  $q_{0j} + q_{1j} + q_{2j} + q_{3j} + q_{4j} = 1$, $\forall j \in \mathbb{N}$ (stochasticity);

2.  $\prod\limits_{j=1}^{\infty} \max\{q_{0j}, q_{1j}, q_{2j}, q_{3j}, q_{4j}\} = 0$ (continuity).

The known theorem [25] states that for any $x \in [0; 1]$ there exists a sequence $(\alpha_k) \in L$ such that

$$x = \beta_{\alpha_1 1} + \sum_{k=2}^{\infty} \left( \beta_{\alpha_k k} \prod_{j=1}^{k-1} q_{\alpha_j j} \right) = \Delta^{Q_5^*}_{\alpha_1 \alpha_2 \dots \alpha_k \dots}, \tag{1}$$

where $\beta_{0j} = 0$, $\beta_{ij} \equiv q_{0j} + q_{1j} + \dots + q_{i-1,j} = \beta_{i-1,j} + q_{i-1,j}$, $i \in \{1, 2, 3, 4\}$, $j \in \mathbb{N}$.

The representation of the number $x$ in the form of series (1) is called $Q_5^*$-expansion while the symbolic notation $x = \Delta^{Q_5^*}_{\alpha_1 \alpha_2 \dots \alpha_k \dots}$ is called $Q_5^*$-representation, and a number $\alpha_k = \alpha_k(x)$ is called j-th digit in the representation of $x$.

If all the columns of the matrix $(q_{ij})$ are identical, i.e., $q_{ij} = q_i$ for any $j \in \mathbb{N}$, then $Q_5^*$-representation is called $Q_5$-representation. If $q_i = \frac{1}{5}$ is for all $i \in A_5$, then the $Q_5$-representation is a classic a five-digit representation.

Each irrational number has a unique representation, but some rational numbers have two representations. These are numbers with the representations $\Delta^{Q_5^*}_{\alpha_1 \alpha_2 \dots \alpha_m (0)} = \Delta^{Q_5^*}_{\alpha_1 \alpha_2 \dots \alpha_{m-1} [\alpha_m - 1](4)}$. By agreement, we use only one of two representations of a rational number containing period (0). Then, we have the uniqueness of the $Q_5^*$-representation of a number.

The concepts of cylinder and Hausdorff–Besicovitch dimension are important for image geometry. We will revisit them [25].

**Definition 1.** *A set of all numbers $x \in [0; 1]$ that have $Q_5^*$-images with the first digits of $c_1, c_2, \dots, c_m$, respectively, is called a cylinder of rank $m$ $\Delta^{Q_5^*}_{s_1 s_2 \dots s_m}$ with base $c_1, c_2, \dots, c_m$.*

The cylinders have the following properties:

(1) Cylinders of rank $m$ are a union of cylinders of rank $m + 1$, i.e.,

$$\Delta^{Q_5^*}_{c_1 c_2 \dots c_m} = \Delta^{Q_5^*}_{c_1 c_2 \dots c_m (0)} \cup \Delta^{Q_5^*}_{c_1 c_2 \dots c_m (1)} \cup \Delta^{Q_5^*}_{c_1 c_2 \dots c_m (2)} \cup \Delta^{Q_5^*}_{c_1 c_2 \dots c_m (3)} \cup \Delta^{Q_5^*}_{c_1 c_2 \dots c_m (4)};$$

(2) A cylinder $\Delta^{Q_5^*}_{s_1 s_2 \dots s_m}$ is a segment with the endpoints

$$a = \Delta^{Q_5^*}_{s_1 s_2 \dots s_m (0)} = \beta_{c_1 1} + \sum_{k=2}^{m} \left( \beta_{c_k k} \prod_{j=1}^{k-1} q_{c_j j} \right), \quad b = \Delta^{Q_5^*}_{s_1 s_2 \dots s_m (4)} = a + \prod_{i=1}^{m} q_{c_i i};$$

(3) Basic metric ratio:

$$\frac{\left| \Delta^{Q_5^*}_{s_1 s_2 \dots s_m (i)} \right|}{\left| \Delta^{Q_5^*}_{s_1 s_2 \dots s_m} \right|} = q_{i, m+1} = const;$$

(4) $\max \Delta^{Q_5^*}_{s_1 s_2 \dots s_m i} = \min \Delta^{Q_5^*}_{s_1 s_2 \dots s_m [i+1]}$, $i = 0, 1, 2, 3$;

(5) For any sequence $(c_n) \in L$, the equation holds:

$$\bigcap_{m=1}^{\infty} \Delta^{Q_5^*}_{s_1 s_2 \dots s_m} = \Delta^{Q_5^*}_{s_1 s_2 \dots s_m \dots} \equiv x \in [0; 1].$$

Let $(M, \rho)$ be a metric space, $E$ a bounded subset of $M$ and $d(E)$ denote the diameter of the set E. Let $\Phi_M$ be a family of subsets of the space $M$ such that for an arbitrary set $E \subset M$ and, for each number $\varepsilon > 0$, there exists an at most countable $\varepsilon$-covering $\{Ej\}$ of $E$ $(E_j \in \Phi_M, d(E_j) \leq \varepsilon)$. Let $\alpha$ be a positive number.

**Definition 2.** *The α-dimensional Hausdorff measure of a bounded set E of a metric space $(M, \rho)$ is defined by*

$$H_\alpha(E) = \lim_{\varepsilon \to 0} \left[ \inf_{d(E_j) \leq \varepsilon} \left\{ \sum_j d^\alpha(E_j) \right\} \right] = \lim_{\varepsilon \to 0} m_\varepsilon^\alpha(E),$$

*where the infimum is taken over all at most countable ε -coverings $\{E_j\}$ of E, $E_j \in \Phi_M$.*

**Definition 3.** *The positive number*

$$\alpha_0(E) = \sup\{\alpha : H_\alpha(E) = \infty\} = \inf\{\alpha : H_\alpha(E) = 0\}$$

*is called the Hausdorff–Besicovitch dimension of the set E.*

Using $Q_5^*$-representation of numbers, denote the function by the equality:

$$
\begin{aligned}
f(x) &= \gamma_{\alpha_1(x)1} + \gamma_{\alpha_2(x)2}g_{\alpha_1(x)1} + \gamma_{\alpha_3(x)3}g_{\alpha_2(x)2}g_{\alpha_1(x)1} + \cdots = \\
&= \gamma_{\alpha_1(x)1} + \sum_{k=2}^{\infty} \left( \gamma_{\alpha_k(x)k} \prod_{j=1}^{k-1} g_{\alpha_j(x)j} \right) \equiv \Delta_{\alpha_1(x)\alpha_2(x)\dots\alpha_k(x)\dots}^{G_5^*}
\end{aligned}
\tag{2}
$$

where $(\overline{g_n}) = (g_{0n}, g_{1n}, g_{2n}, g_{3n}, g_{4n})$, $n \in \mathbb{N}$ is a sequence of vectors such that:

$$
\begin{aligned}
&g_{0n} = \tfrac{\varepsilon_n+2}{4}, \ g_{1n} = -\tfrac{\varepsilon_n}{4}, \ g_{2n} = 0, \ g_{3n} = -\tfrac{\varepsilon_n}{4}, \ g_{4n} = \tfrac{\varepsilon_n+2}{4}, \\
&\gamma_{0n} = 0, \ \gamma_{1n} = \tfrac{\varepsilon_n+2}{4}, \ \gamma_{2n} = \tfrac{1}{2} = \gamma_{3n}, \ \gamma_{4n} = \tfrac{2-\varepsilon_n}{4}, \\
&\text{where } \gamma_{i+1,n} = \gamma_{in} + g_{in}, \ i \in \mathbb{N},
\end{aligned}
\tag{3}
$$

$(\varepsilon_n)$ is a sequence of positive real numbers with $0 \leq \varepsilon_n \leq 1$.
Function properties [29]:

(1) The function is continuous on [0; 1] and acquires all its values from [0; 1];
(2) The function has no intervals of monotonicity, except for intervals of constancy, if $\varepsilon_n \neq 0$ is satisfied for an infinite set of values of *n*;
(3) is a function of limited variation if $\sum\limits_{k=1}^{\infty} \varepsilon_k < \infty$;
(4) is a singular Cantor-type function with a Hausdorff–Besicovitch fractal dimension $\log_5 4$;
(5) the graph of the function is symmetric about point $\left( \frac{1}{2}; \frac{1}{2} \right)$.

## 3. Results

We describe the method of fractal encoding of images using this class of nonmonotonic singular functions. The image is placed in a single square. We define two vectors Q and G such that:

$$Q = \{q_{0j}, q_{1j}, q_{2j}, q_{3j}, q_{4j}\}, \ q_{ij} > 0, \ q_{0j} + q_{1j} + q_{2j} + q_{3j} + q_{4j} = 1, \ j \in N,$$
$$G = \{g_{0k}, g_{1k}, g_{2k}, g_{3k}, g_{4k}\}, \ g_{ik} > 0, \ g_{0k} + g_{1k} + g_{2k} + g_{3k} + g_{4k} = 1, \ k \in N.$$

The first vector divides our image along the abscissa axis into five Q-cylinders (rank areas) that do not intersect (Figure 6), the length of $q_{0j}, q_{1j}, q_{2j}, q_{3j}, q_{4j}$ respectively. The second vector on the y-axis specifies a set of G-cylinders that can overlap (Figure 7), the length of $g_{0k}, g_{1k}, g_{2k}, q_{3k}, q_{4k}$ each, respectively.

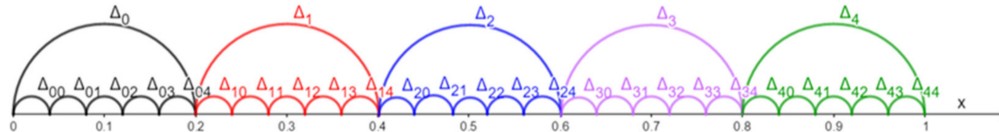

**Figure 6.** Q-cylinders of the 1st and 2nd rank.

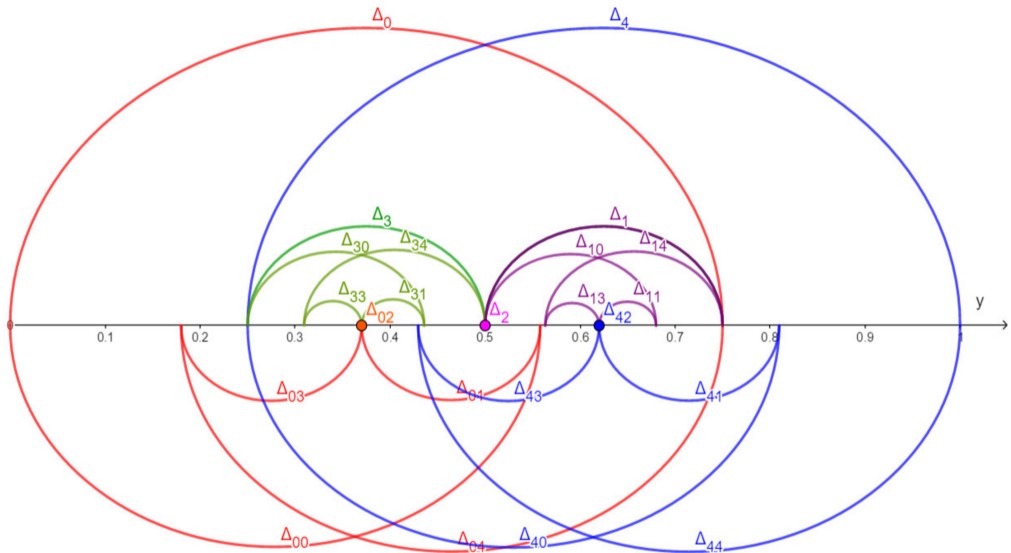

**Figure 7.** G-cylinders of the 1st and 2nd rank.

We take two identical images of $1 \times 1$ size. The first image is divided into Q-cylinders, and the second image is divided into G-cylinders (they describe similar parts and are used in constructed decoded images). For each method of search of Q-cylinders, the nearest G-cylinder for which the distributed features can be approximated by the distribution of a rank area is selected. For the best approach to the G-cylinders, use the official conversion, which helps to change the brightness and contrast. If the desired approximation is not achieved, each cylinder of the rank area again extends to the appearance of the corresponding parts and the process is repeated. The numbers of the Q- and G-cylinders that were used in the encoding process and helped to obtain the desired results, together with the coefficients of affine transformations, are written to the file. These results will then be used in decoding.

Therefore, in the original image there is a search for self-similar areas, which are then taken as the basic elements of the fractal image. The latter is approximated by fractal transformations, and then we obtain an image in the form of Formula (2), which reflects the transformation.

$$T_{0n}(x, y) = (q_{0n}x; g_{0n}),$$

$$T_{mn}(x, y) = \left( q_{mn}x + \sum_{i=0}^{m-1} q_{in}; g_{mn}y + \sum_{i=0}^{m-1} g_{in} \right), \ n \in \mathbb{N}, \ m = 1, 2, 3, 4.$$

**Theorem 1.** *The iterated function system defines a single set F such that* $F = \bigcup\limits_{i=0}^{4} T_{in}(F)$, $n \in \mathbb{N}$ [30].

$T_{in}$ is a compression image, so the $\{T_{0n}, T_{1n}, T_{2n}, T_{3n}, T_{4n}\}$ family is an iterated function system. We show that the set $F$ is a graph of a function continuous on $[0; 1]$. We give a

geometric interpretation of the construction of this set. Let the graph of the function $F_0(x)$ be broken, connecting series points:

$$(0;0), (q_{01}; g_{01}), \left( \sum_{i=0}^{1} q_{i1}; \sum_{i=0}^{1} g_{i1} \right), \left( \sum_{i=0}^{2} q_{i1}; \sum_{i=0}^{2} g_{i1} \right), \left( \sum_{i=0}^{3} q_{i1}; \sum_{i=0}^{3} g_{i1} \right),$$

$$\left( \sum_{i=0}^{4} q_{i1}; \sum_{i=0}^{4} g_{i1} \right) = (1;1).$$

Graph of the function $F_1(x)$ is broken, connecting series points:

$$(0;0), (q_{01}q_{02}; g_{01}g_{02}), \left( q_{01} \sum_{i=0}^{1} q_{i2}; g_{01} \sum_{i=0}^{1} g_{i2} \right), \left( q_{01} \sum_{i=0}^{2} q_{i2}; g_{01} \sum_{i=0}^{2} g_{i2} \right),$$

$$\left( q_{01} \sum_{i=0}^{3} q_{i2}; g_{01} \sum_{i=0}^{3} g_{i2} \right), \left( q_{01} \sum_{i=0}^{4} q_{i2}; g_{01} \sum_{i=0}^{4} q_{i2} \right) = (q_{01}; g_{01});$$

$$(q_{01} + q_{11}q_{02}; g_{01} + g_{11}g_{02}), \left( q_{01} + q_{11} \sum_{i=0}^{1} q_{i2}; g_{01} + g_{11} \sum_{i=0}^{1} g_{i2} \right),$$

$$\left( q_{01} + q_{11} \sum_{i=0}^{2} q_{i2}; g_{01} + g_{11} \sum_{i=0}^{2} g_{i2} \right), \left( q_{01} + q_{11} \sum_{i=0}^{3} q_{i2}; g_{01} + g_{11} \sum_{i=0}^{3} g_{i2} \right),$$

$$\left( q_{01} + q_{11} \sum_{i=0}^{4} q_{i2}; g_{01} + g_{11} \sum_{i=0}^{4} q_{i2} \right) = \left( \sum_{i=0}^{1} q_{i1}; \sum_{i=0}^{1} g_{i1} \right);$$

$$\left( \sum_{i=0}^{2} q_{i1} + q_{31}q_{02}; \sum_{i=0}^{2} g_{i1} + g_{31}g_{02} \right), \left( \sum_{i=0}^{2} q_{i1} + q_{31} \sum_{i=0}^{1} q_{i2}; \sum_{i=0}^{2} g_{i1} + g_{31} \sum_{i=0}^{1} g_{i2} \right),$$

$$\left( \sum_{i=0}^{2} q_{i1} + q_{31} \sum_{i=0}^{2} q_{i2}; \sum_{i=0}^{2} g_{i1} + g_{31} \sum_{i=0}^{2} g_{i2} \right),$$

$$\left( \sum_{i=0}^{2} q_{i1} + q_{31} \sum_{i=0}^{3} q_{i2}; \sum_{i=0}^{2} g_{i1} + g_{31} \sum_{i=0}^{3} g_{i2} \right),$$

$$\left( \sum_{i=0}^{2} q_{i1} + q_{31} \sum_{i=0}^{4} q_{i2}; \sum_{i=0}^{2} q_{i1} + g_{31} \sum_{i=0}^{4} q_{i2} \right) = \left( \sum_{i=0}^{3} q_{i1}; \sum_{i=0}^{3} g_{i1} \right);$$

$$\left( \sum_{i=0}^{3} q_{i1} + q_{41}q_{02}; \sum_{i=0}^{3} g_{i1} + g_{41}g_{02} \right), \left( \sum_{i=0}^{3} q_{i1} + q_{41} \sum_{i=0}^{1} q_{i2}; \sum_{i=0}^{3} g_{i1} + g_{41} \sum_{i=0}^{1} g_{i2} \right),$$

$$\left( \sum_{i=0}^{3} q_{i1} + q_{41} \sum_{i=0}^{2} q_{i2}; \sum_{i=0}^{3} g_{i1} + g_{41} \sum_{i=0}^{2} g_{i2} \right),$$

$$\left( \sum_{i=0}^{3} q_{i1} + q_{41} \sum_{i=0}^{3} q_{i2}; \sum_{i=0}^{3} g_{i1} + g_{41} \sum_{i=0}^{3} g_{i2} \right),$$

$$\left( \sum_{i=0}^{3} q_{i1} + q_{41} \sum_{i=0}^{4} q_{i2}; \sum_{i=0}^{3} q_{i1} + g_{41} \sum_{i=0}^{4} q_{i2} \right) = (1;1),$$

i.e., $F_1(x) = \bigcup_{i=0}^{4} T_i(F_0(x))$. These points are uniquely determined by the vectors $Q$ and $G$, and they belong to the interior of the square $[0;1] \times [0;1]$. We say that the transformation $T$ is performed over the segments of the broken $F_0(x)$. With each of the segments of the obtained broken $F_1(x)$, which are not segments of constancy, we do the same (we perform the transformation $T$ on them). Continuing this process, we obtain a functional sequence $(F_n(x))$ such that:

$$F_n(x) = T(F_{n-1}(x)) = \bigcup_{i=0}^{4} T_i(F_{n-1}(x)).$$

Thus, according to Banach's theorem (the theorem was formulated and proved in 1922 by Stefan Banach and is one of the most classical and fundamental theorems of functional analysis), there is a class of mappings—these are compressive mappings. A well-known

statement: if we repeatedly apply the map $T$ to the image $F_0$ so that $F_i = T(F_{i-1})$, then for $i \to \infty$ we get the same image, regardless of the initial $F_0$:

$$F = \lim_{i \to \infty} F_i.$$

The image $F$ is called the *original image* of the transformation $T$.

Let $q_{ij} = q_i = \frac{1}{5}$ and $\varepsilon_n = 1$. Then,

$$Q = \left\{ \frac{1}{5}; \frac{1}{5}; \frac{1}{5}; \frac{1}{5}; \frac{1}{5} \right\}, \quad G = \left\{ \frac{3}{4}; -\frac{1}{4}; 0; -\frac{1}{4}; \frac{3}{4} \right\}.$$

The graph of the function is constructed according to the following algorithm. First, we build a unit square as a basis. We fix points $(0;0)$, $\left( \frac{1}{5}; \frac{3}{4} \right), \left( \frac{2}{5}; \frac{2}{4} \right), \left( \frac{3}{5}; \frac{2}{4} \right), \left( \frac{4}{5}; \frac{1}{4} \right)$, $(1;1)$ according to the coordinates of the vector $\left( \vec{g_1} \right) = \left\{ \frac{3}{4}; -\frac{1}{4}; 0; -\frac{1}{4}; \frac{3}{4} \right\}$ and connect them broken (Figure 8). These points and the segment of constancy belong to the graph of the function. Next, around the other four broken we build rectangles, considering them diagonals. In each of these rectangles we continue to construct new broken lines according to the coordinates of the vector $\left( \vec{g_2} \right) = \left\{ \frac{3}{4}; -\frac{1}{4}; 0; -\frac{1}{4}; \frac{3}{4} \right\}$ (Figure 9). In the next step, the algorithm is repeated, i.e., each of the four broken lines is replaced by five new broken lines at the appropriate scale. The more iterations you make, the more accurate the graph of the function will be (Figure 10).

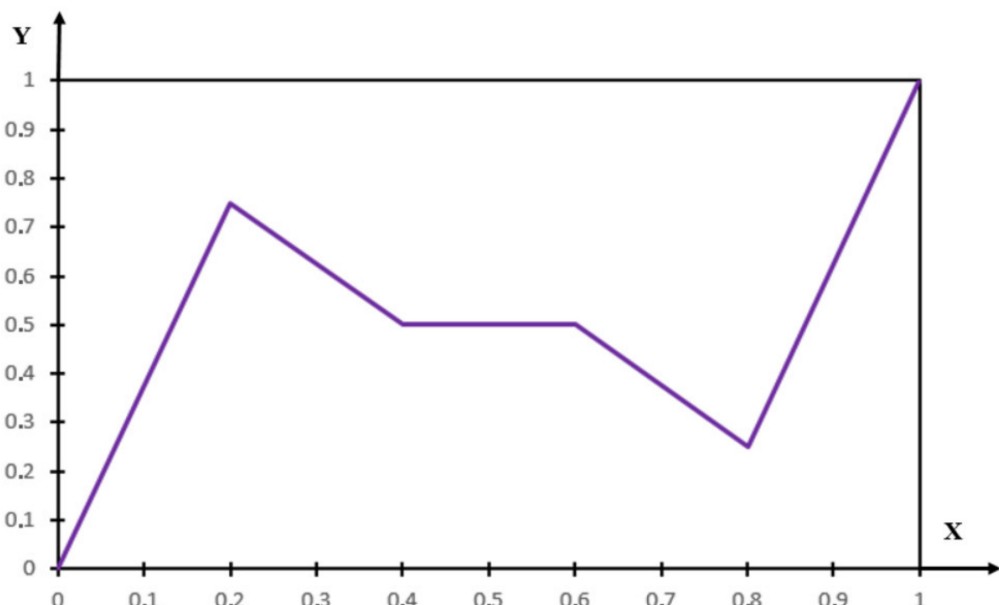

**Figure 8.** Graph of function $F_0(x)$.

Increasing iterations is responsible for the accuracy of the graph.

Function (2) has an interesting property—if we know the initial sets of numbers, it is easy to calculate the value of the function, but, conversely, when the values of the function are known, the initial set of numbers is difficult to recover because there are many such sets.

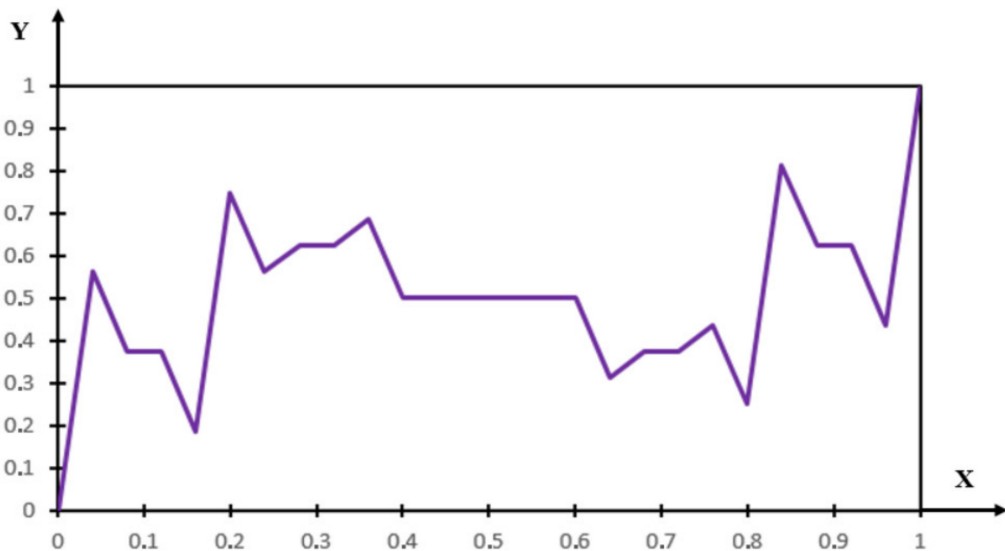

**Figure 9.** Graph of function $F_1(x)$.

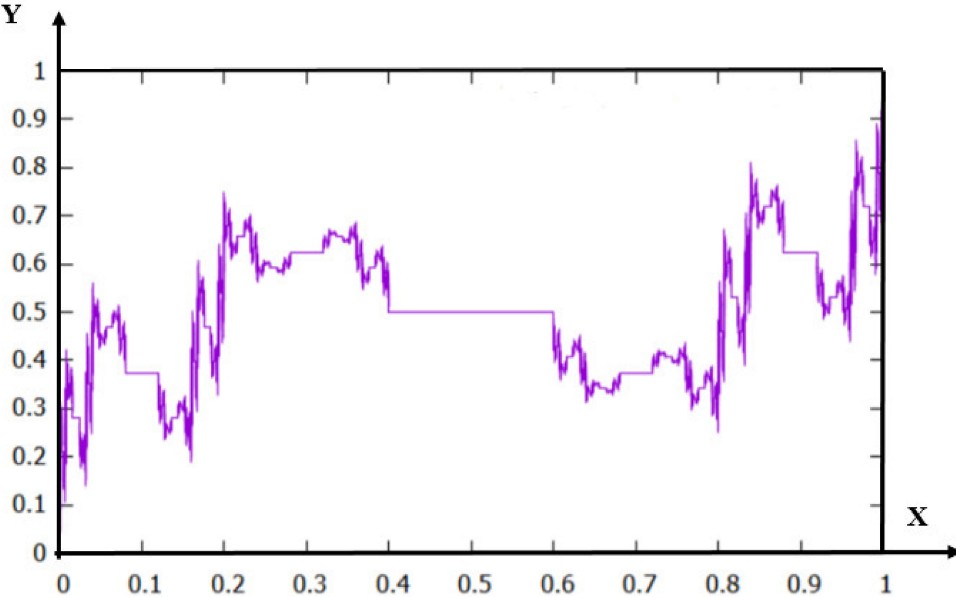

**Figure 10.** Graph of function $F_{15}(x)$.

Another special property of this function is the manifestation of self-similar properties of the graph of the function depending on the parameter at intervals where the function is not constant, i.e., the graph of the function $\Gamma_f = \{(x, f(x)), x \in [0,1]\}$ is a self-similar set and

$$\Gamma = \underset{i \in A_5, i \neq 2}{\cup} \varphi_i(\Gamma), \; where \; \varphi_k(\Gamma) \neq \varphi_p(\Gamma), \; k \neq p,$$

$$\varphi_i : \begin{cases} x' = \frac{1}{5}x + \frac{i}{5}, \\ y' = g_{i1}y + \gamma_{i1} \end{cases}.$$

This allows for encoding information faster, thus increasing the efficiency of data transmissions through communication channels.

The process of encoding information requires a lot of calculations. Large volumes of iterations are performed to search for self-similar fragments in the image. Therefore, compressing a single image takes a long time. In this case, the more iterations there are to make, the more accurate the result will be.

Decoding a fractal image is also an iterative process, although it takes little time, as all such objects are searched for in the encoding process. All you need to do is refine the fractal codes by transforming them into the original image. However, if you do not know the image encoding algorithm, the decoding process will be very cumbersome and time consuming.

Here are some examples of fractal encoding with given initial sets of digits for a given image $f$:

1. For Q-cylinders, the digits 0 and 1 are allowed, i.e., the image will be divided into cylinders $\Delta_0, \Delta_1, \Delta_{00}, \Delta_{01}, \Delta_{10}, \Delta_{11}, \ldots$. Then, for the second identical image, as a result of affine transformations, the brightness distribution will contain G-cylinders with numbers 0 and 1, i.e., the image of the set $C_1 \equiv C[Q_5^*; \{0,1\}]$ is the segment $\left[0; \frac{3}{4}\right]$ (Figure 11);

2. For Q-cylinders, the digits 1 and 3 are allowed, i.e., the image will be divided into cylinders $\Delta_1, \Delta_3, \Delta_{11}, \Delta_{13}, \Delta_{31}, \Delta_{33}, \ldots$. Then, for the second identical image, as a result of affine transformations, the brightness distribution will contain G-cylinders with numbers 1 and 3, i.e., the digits of the set $C_2 \equiv C[Q_5^*; \{1,3\}]$ are a set of Cantor type $C_3 \equiv C[G_5^*; \{1,3\}]$ (it is a set of not deleted points; it is possible to define the relation of this set to unit interval through the general length of the removed subintervals) (Figure 12);

3. For Q-cylinders, the digits 1,2 and 3 are allowed, i.e., the image will be divided into cylinders $\Delta_1, \Delta_2, \Delta_3, \Delta_{11}, \Delta_{12}, \Delta_{13}, \Delta_{21}, \Delta_{22}, \Delta_{23}, \Delta_{31}, \Delta_{32}, \Delta_{33}, \ldots$. Then, for the second identical image, as a result of affine transformations, the brightness distribution will contain G-cylinders with digits 1,2 and 3, i.e., the image $C_4 \equiv C[Q_5^*; \{1,2,3\}]$ is the set $E = C_3 \cup M$ (Figure 13), where $C_3$ is a set of Cantor type and M is a discrete subset of the set of five-rational numbers:

$$M = \left\{ y: \ y = \Delta_{\alpha_1 \alpha_2 \ldots \alpha_{m-1} 2(0)}^{G_5^*}, \ \alpha_i \in \{1,3\}, \ m \in \mathbb{N} \right\};$$

4. For Q-cylinders, the digits 0 and 4 or 1 and 3 are allowed. Then, for the second identical image, as a result of affine transformations, the brightness distribution will contain G-cylinders with digits 1,2,3,4, if the digits 0 and 4 or 1 and 3 are allowed for Q-cylinders, then the image of the set of Cantor type $C_5 \equiv C[G_5^*; V_n]$ (Figure 14), where

$$V_n = \begin{cases} \{0,4\}, & if \ n = 1 \pmod 3, \\ \{1,3\}, & if \ n \neq 1 \pmod 3, \end{cases}$$

is a set of Cantor type of Lebesgue zero measure.

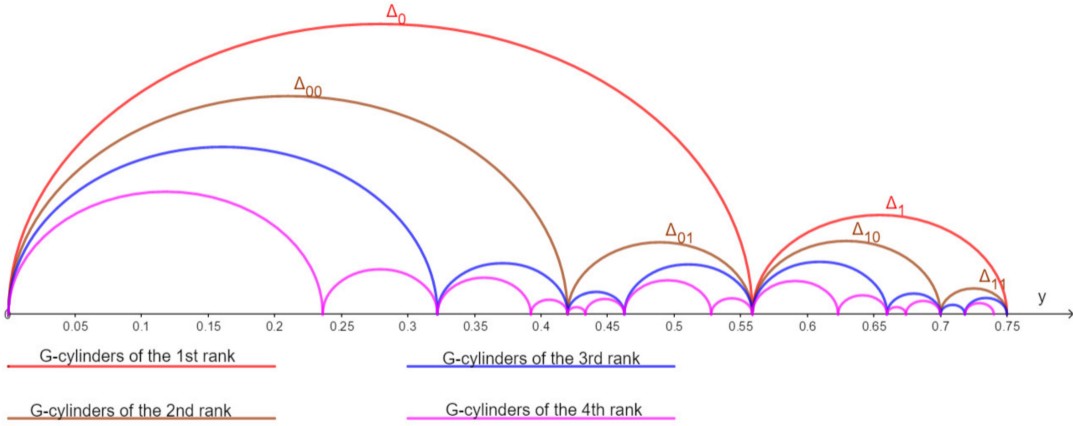

**Figure 11.** The image of the plural $C_1$.

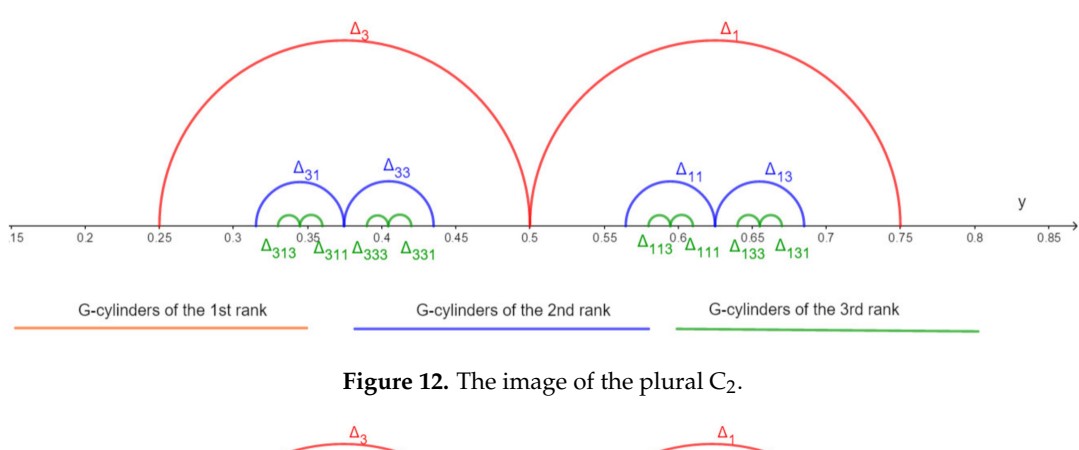

**Figure 12.** The image of the plural $C_2$.

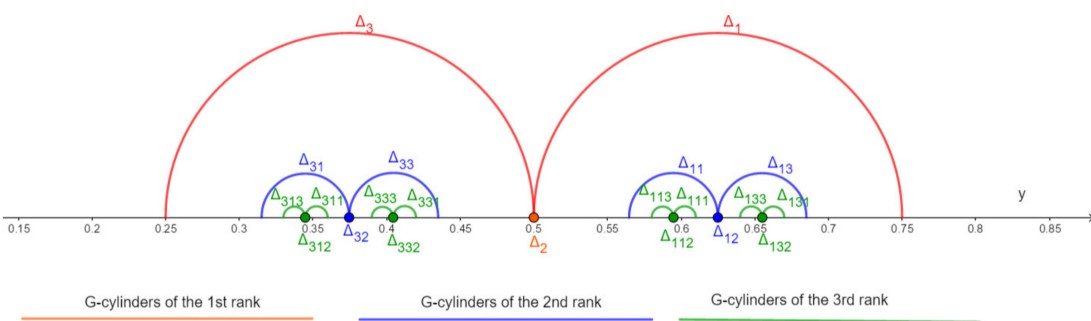

**Figure 13.** The image of the plural $C_4$.

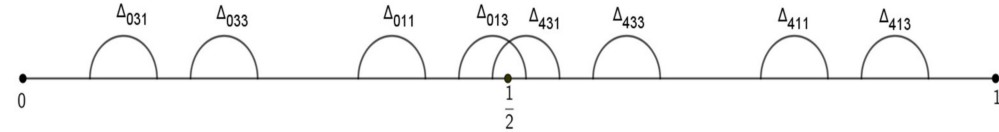

**Figure 14.** The image of the plural $C_5$: G-cylinders of the 3rd rank.

This function class model allows for the development of a new fractal image encoding method for data transmission over communication channels, storage on media and subsequent use in geographic information systems.

The obtained results can be effectively used for the computer processing of aerial photographs in GIS of various functional purposes. A promising development of this mathematical model is the consideration of the problems of its practical application in GIS of environmental monitoring for compression and archiving of geospatial data.

## 4. Discussion

This article highlights a new method of image compression, which uses a class of nonmonotonic singular functions with fractal properties and depends on a finite number of parameters. These features allow you to get excellent digital data compression ratios and fast decoding. Another feature is that the size of the data after compression will actually take up less space in the file. This makes it possible to transfer information from satellites to Earth faster and then use it in geospatial systems, for example, for weather forecasting, studying Earth resources, and so on.

The disadvantage of this method is the large amount of computation, because to obtain a high degree of compression, you need to perform a large number of conversions, which can degrade image quality. The data we will receive after unpacking may differ from the initial ones, but the degree of difference will not be significant with their further use.

Unfortunately, the proposed methods do not completely solve the problem of increasing the speed and efficiency of fractal compression. However, today, more and more scientists are trying to improve the efficiency of existing methods and find new ways to optimize fractal encoding.

## 5. Conclusions

Mathematical models of the effective use of fractal functions for compression (encoding) of raster images are covered. There are already a large number of existing models using functions with a complex local structure, but they also need to be improved. Functions with fractal properties, unlike conventional functions, help to efficiently encode data and solve complex problems in various areas of human activity. Such functions are given by a recursive formula. Their generation takes a long time, which provides a high degree of compression, but is time consuming. Unpacking the image is easier, because the main work has already been done during encoding and it remains only with the help of known fractal codes to return the raster image. The obtained results allow one to create a sufficiently reliable mathematical support for the process of compression of various graphic information and to improve existing methods.

We see prospects for further research in constructing a family of functions with fractal properties, using different systems of encoding real numbers, and their application to create reliable and advanced methods of encoding spatial data, their storage, processing and representation. This will create a single information space and provide ample opportunities for systematic analysis of information for effective environmental quality management and ensuring the safety of life.

**Author Contributions:** Conceptualization, O.S. and O.L.; methodology, O.S., O.L. and O.B.; validation, O.B. and O.L.; formal analysis, J.N. and R.K.; investigation, J.N. and R.K.; writing—original draft preparation, O.B. and R.K.; writing—review and editing, O.S. and O.L. All authors have read and agreed to the published version of the manuscript.

**Funding:** This research received no external funding.

**Institutional Review Board Statement:** Not applicable.

**Informed Consent Statement:** Not applicable.

**Acknowledgments:** The authors would like to thank the editor and the referees for his/her careful reading and valuable comments.

**Conflicts of Interest:** The authors declare that there is no conflict of interest.

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
