# Peer review of "Image Compression Using Fractal Functions"

_fractalfract, doi:10.3390/fractalfract5020031_

Round 1

Reviewer 1 Report

The use of fractal functions for processing of images, which were transmitted via satellite radio channel of spacecraft, is considered. Specifically, a method for constructing image encoding-decoding by using a class of continuous functions that depend on a finite set of parameters and have fractal properties is investigated. The authors use a coding system with a finite alphabet to build a new mathematical model that will be used in fractal image compression. The model is based on a continuous class of continuous functions that depend on a finite set of parameters and have fractal properties.

In some cases, the text is very unclear and the reader has to guess the exact meaning. For instance, what does the author mean by complex problems? The authors also use inappropriate terms and expressions, for instance system of iterative functions instead of iterated function systems, coding instead of encoding, as in the phrase “Fractal image compression (fractal transformation, fractal coding) is a lossy image compression algorithm based on a system of iterated functions to images”, etc; for an explanation see [1]. Moreover, the text contains some errors in syntax, grammar and vocabulary.

Although the article highlights a new method of image compression which uses a class of nonmonotonic singular functions with fractal properties and depend on a finite number of parameters, the proposed methods do not completely solve the problem of increasing the speed and efficiency of fractal-based image compression; see [2].

Finally, the authors, together with the article, must submit the computer program as well as the original photos used to produce the compressed ones. I’m afraid I have to suggest rejection of the paper in this form.

  1. Drakopoulos V., Fractal-based image encoding and compression techniques, – Scientific Letters of the University of Žilina 15 (3) (2013), 48–55.
  2. Rafik Menassel (July 28th 2020). Optimization of Fractal Image Compression, Fractal Analysis - Selected Examples, Robert Koprowski, IntechOpen, DOI: 10.5772/intechopen.93051. Available from: https://www.intechopen.com/books/fractal-analysis-selected-examples/optimization-of-fractal-image-compression

Author Response

Dear reviewer,
Thank you very much for taking the time to review our article. All your recommendations are taken into account.
Your recommendations only improved our article.

Sincerely, Dr. Laptev

Reviewer 2 Report

Mathematical description may be not clear for normal reader. In the text are some mistakes, e.g.

  • some texts are in Cyrillic (not in Latin), e.g. row 290,291, ...
  • Figure 5 must be translate to English
  • in Figure 6 missing the indexes in last circle (5)
  • in Fig. 11 - Fig. 14 is used symbol "delta", must be I think "g"
  • 9 and Fig. 10 - wrong quality of figure, missing description of axis, ...
  • description of Figures 11 - Fig 14 is not clear, there are not evident all parameters used for their construction - must be fulfilled
  • I do not know if the coefficient "gamma" and "g" at row 160 are in order
  • I think that text at row 177 not correspond with text in Fig. 7 - coordinates must be "g" not "q"
  • It must be detailed discuss "zero" cycles (cycles with radius equal to zero)
  • row 123 not correlate row 127 (5 dimensional matrix, N-dimensional matrix, infinite dimensional matrix)
  • row 148, row 152 - missing brackets (0), (1), ...
  • I od not understand what is mean by definition at row 270, row 276, row. 277, row 282 and row 289 and how that correlate with Figures 11 - 14. I think that for construction of these figures are necessary more parameters, not only these equations

Conclusion:

text must be carefully reread and must be repaired mistakes and fulfill the comment to better understand of subject.

Author Response

(The authors gave the same response as above.)

Reviewer 3 Report

In this paper, the authors provide the mathematical foundations of a novel procedure for image compression which involves a class of fractal functions and only depends on a finite set of parameters. They state that their approach allows obtaining excellent compression ratios for digital images and a fast decoding, as well. A wide range of applications would follow from their encoding proposal. 

In my opinion, the content of the paper is interesting, though there are several issues that should be clarified before considering it for acceptance in Fractal and Fractional. Regarding them, next I provide a list of comments, questions, and suggestions that should be addressed by the authors.

Replace earth by Earth along the whole manuscript.

Fig. 1: What role does the scanner (?) play in the figure at the bottom of that picture?

line 69: consists in <--> consists of

line 71: In [5], ...

line 74: ``Jacqueline's method'' should be replaced by ``Jacquin's method'', instead.

line 75: Fischer should be Fisher.

line 100: ... when being enlarged repeat accurately a larger figure.

line 101: what should be understood by fractional infinite dimension?

line 103: Replace ``coasts of seas and oceans'' by coastlines or contours of coasts.

Fig. 4: captions regarding levels are labeled in Spanish.

Fig. 5 Should be centered.

line 104: add ``and'' before the last example.

line 112: the plural form of basis is bases.

line 114: Sierpi\'nski is not well written. Revise also the details of reference [24].

line 119: depends.

line 122: Why not writing L=A_5^{\mathbb{N}}?

line 123: N <--> \mathbb{N}. See also lines 176, 177, 202, 205, 284, etc., in this regard. 
Also, A_S should be A_5. Revise also lines 176 and 177, as well.

lines 123 and 124: I would replace the notation \|q_{ij}\| by (q_{ij})_{i\in A_5, j\in \mathbb{N}}, instead. The symbol \|\cdot \| is usually associated to the concept of norm.

line 130: revise Eq. (1) since it is not clear in what range the index j varies. 

line 132: what does it mean the overline symbol that appears therein? Further, the Cyrillic characters apeared therein should be replace by ``with'' (or something like that).

line 160: the Cyrillic characters apeared therein should be replace by ``where''.

line 167: By Hausdorff-Besicovitch dimension do you mean box dimension? Also, since your enconding approach is proposed on the basis of sngular functions, I think that such a concept should be defined somewhere.

line 183 (caption of Fig. 6): Q-cylinders.

line 185 (caption of Fig. 7): G-cylinders.

lines 188-191: please, provide additional details regarding the connection between brightness (resp., contrast) of a given image and looking for appropriaye G-cylinders (resp., Q-cylinders).

The result appeared in line 205 should be stated as a theorem.

What about the legend of Fig. 10?

lines 253 and 255: revise these mathematical expressions. What should we understand by A, \alpha_1, or f?  \Gamma_i=Q_i? Please, explain.

line 290: the Cyrillic characters apeared therein should be replace by ``if''.

Figure 14 should appear centered.

line 295: in what sense is the proposed methodology ``efficient''?

Regarding figures 8,9, and 10: the format of such pictures should be the same in the manuscript.

Take a look at the references. Some of them contain typos (e.g., in [1] should appear ``Barnsley''). Refs. [18] and [19] due to (Benoît) Mandelbrot appear referenced from Russian sources. Maybe they should be replaced by the original sources, instead. 

I also suggest them to revise the English drafting along the whole manuscript.

Author Response

Dear reviewer!
Mistakes corrected.
Thanks for the comments, they will allow us to make the article better
Sincerely, Dr. Laptiev

Round 2

Reviewer 1 Report

Without illustrating original photographs or images compared and contrasted with the compressed or encoded ones, I do not feel that the article is publishable in any form. Moreover, there is no "de-encoding" within the scientific literature.

Author Response

Dear reviewer!
Apparently, there is a small misunderstanding.
The article is theoretical. University software was used.  That is, there is no new program that could compress images with fractals.
Writing a program is a purely practical aspect based on scientific developments and regulations.
We have tried to correct the shortcomings you mentioned.
We look forward to mutual understanding.
Sincerely, Dr. Laptev

Reviewer 2 Report

I have only two comments to this paper now:

  • on the row 167 must be Hausdorf (not Gausdorf)
  • Figures 8-10 must be better quality. I think that it will be proper, the curves set to one Figure.
  • missing axis at these Figures

Author Response

1) on the row 179 must be Hausdorf (not Gausdorf) –  fixed

2) Figures 8-10 must be of better quality. I think that it will be proper, the curves set to one Figure – it is not advisable to draw graphs on one field (for example, as for Figures 4-5)

3) Missing axis at these Figures 8-10 – axes of figures signed

Reviewer 3 Report

The authors have implemented the changes I suggested, thus improving the first version of their manuscript. However, there are still minor details that should also be addressed. In this way, I suggest them taking into account the next suggestions. Please, note that the line numbers appearing below correspond to the last version of their manuscript.

line 106: Sierpi\'nski's triangle.

line 167: Hausdorff-Besicovitch dimension, i.e., box dimension. By merely writing this, you can skip the explication provided into parentheses.

line 187: ``and the second image is divided into G-cylinders ...''

line 234 should not begin by a comma.

line 254: ``..., i.e., the garph of the function \Gamma_f=[...] is a self-similar set ...''

line 258: ``This allows enconding information faster, thus increasing the efficiency of data transmissions through communication channels.''

Author Response

line 106, 112, 419: Sierpi\'nski's triangle. – ''Serpinsky'' has been fixed on ''Sierpinski''

line 415, 416, 419,420,421,423 – in addition, errors in the bibliography were identified and corrected.

line 167: Hausdorff-Besicovitch dimension, i.e., box dimension. By merely writing this, you can skip the explication provided into parentheses. – an explanation is given Hausdorff-Besicovitch dimension in lines 142-143, 157-167.

line 197: ``and the second image is divided into G-cylinders ...'' – letter ''G'' added.

line 244 should not begin by a comma – fixed.

line 264: ``..., i.e., the garph of the function \Gamma_f=[...] is a self-similar set ...'' – grammatical errors corrected.

lines 284, 291, 301 – additionally corrected errors in the abbreviation '' i.e.''.

lines 268-269: ``This allows encoding information faster, thus increasing the efficiency of data transmissions through communication channels.'' – grammatical errors in the sentence are corrected.
